# DATA-CENTRIC GRAPH CONDENSATION VIA DIFFUSION TRAJECTORY MATCHING

## ABSTRACT

This paper introduces Data Centric Graph Condensation (named DCGC), a data-centric and model-agnostic method for condensing a large graph into a smaller one by matching the distribution between two graphs. DCGC defines the distribution of a graph as the trajectories of its node signals (such as node features and node labels) induced by a diffusion process over the geometric structure, which accommodates multi-order structural information. Built upon this, DCGC compresses the topological knowledge of the original graph into the orders-of-magnitude smaller synthetic one by aligning their distributions in input space. Compared with existing methods that stick to particular GNN architectures and require solving complicated optimization, DCGC can be flexibly applied for arbitrary off-the-shelf GNNs and achieve graph condensation with a much faster speed. Apart from the cross-architecture generalization ability and training efficiency, experiments demonstrate that DCGC yields consistently superior performance than existing methods on datasets with varying scales and condensation ratios.

## 1 INTRODUCTION

Graphs are a generic representation for systems of certain interactions and structures, such as large online social networks (Fan et al., 2019), user-item recommender systems (Wu et al., 2019), chemical molecules (Stärk et al., 2022), and biological protein interactions (Réau et al., 2023). Recent advances in deep learning-based methods on graph-structured data, such as graph neural networks (Kipf & Welling, 2017; Velickovic et al., 2018), have garnered significant attention and research interest. However, training deep graph networks on large real-world graphs requires tremendous computational and infrastructural resources due to the necessity of performing message passing layer by layer among inter-connected nodes (Zeng et al., 2020).

To address this challenge, a natural idea is to compress the dataset involving data structures, which, in particular, entails reducing the number of nodes and edges in the graph. To this end, traditional methods include graph sparsification (Spielman & Teng, 2011) and graph coarsening (Loukas & Vandergheynst, 2018; Cai et al., 2021): the former aims to obtain a sparser graph by removing edges from the original graph, while the latter targets reducing the number of nodes by extracting a subset from the node-set. However, these methods often rely on some predefined heuristics and lack guidance from training (Yang et al., 2023), making it difficult to achieve satisfactory results on downstream tasks.

**Existing Works.** Another technical path showing empirical success in recent studies resorts to a synthesis-based approach that directly learns the node feature matrix and the adjacency matrix of the target compressed graph (a.k.a. synthetic graph), which is called *graph condensation* or *graph distillation* (Jin et al., 2022; Liu et al., 2022; Yang et al., 2023; Zheng et al., 2023). These methods share a similar spirit, aiming to learn a synthetic graph that can replicate the same gradient trajectory of model parameters as the original graph, named gradient-matching (Zhao et al., 2021). Although these methods have achieved promising performance, their design philosophies lead to unsatisfactory capabilities (Gupta et al., 2024) due to failure to fulfill the following criteria:

1. **Efficiency of condensation process.** The primary goal of graph condensation is to circumvent the need for training on the original large graph, which would be memory and time-inefficient. However, current methods based on gradient-matching require training GNN models on the original full graph,

which contradicts the fundamental objective of graph condensation. In addition, existing works are formulated as a bi-level optimization problem, which further exacerbates the computational cost.

2. **Independence from models and hyper-parameters.** Existing methods based on gradient matching are model-centric and rely on specific GNN architectures and the associated hyper-parameters. This means that any change in the GNN architecture (such as switching from GCN to GAT) or even any change in hyper-parameters (like the number of GNN layers) can lead to significant change in the condensed graph, necessitating a new round of condensation.

3. **Cross-architecture generalization ability.** The stickiness to particular GNNs means that synthetic graphs condensed by one GNN may not adapt well to training with other GNN architectures, a problem we refer to as poor cross-architecture generalization ability. More seriously, synthetic graphs condensed by an inappropriate GNN can lead to a significant performance drop, even when using the appropriate GNN model for training and testing on the condensed graph. This issue is particularly evident in heterophilic graphs.

**Presented Work.** To address these limitations, this paper proposes **D**ata-**C**entric **G**raph **C**ondensation via Diffusion Trajectories Matching (DCGC in short). DCGC inherits the spirit of distribution matching (Zhao & Bilen, 2023), learning the condensed graph by minimizing the divergence between the distributions of the original graph and the synthetic graph. Observing that a graph is a mixture of the node signals (e.g., node features and labels) and their connections, we seek a principled means to characterize and extract the topological knowledge from the original graph entangled with these node signals. In particular, we resort to an analogy between a geometric diffusion process that updates node signals through time and a non-parametric propagation on graphs that returns aggregated node features at different layers. On top of this, we decompose a graph into a collection of node signals, where each node's signal is aggregated from its multi-order structural information, and the distribution of a graph is subsequently defined as the distribution of the aggregated node signals. The divergence between the original graph and the synthetic graph is further measured by the Maximum Mean Discrepancy, which can be easily optimized in linear time w.r.t. the graph size.

DCGC addresses the limitations of the above works in the following ways: 1) Operated in the data space and without relying on training a GNN on the full graph, DCGC eliminates the burdensome bi-level optimization and, therefore, has a much faster training speed. 2) DCGC's condensation process is unsupervised, so it does not rely on tuning the hyper-parameters of a specific GNN for the node classification task. 3) The data-centric property frees DCGC from specifying a GNN architecture for condensation, therefore endowing DCGC with desired cross-architecture generalization capability. The consideration of the label's distribution over the graph structure also enables DCGC to adapt to heterophilic graphs easily.

We evaluate the proposed DCGC on eight graph datasets of varying scales and properties. The experimental results demonstrate that the synthetic graphs condensed by DCGC yield comparable or even better performance than existing SOTA gradient-matching methods. In cross-architecture settings and on heterophilic datasets, DCGC exhibits superior and more stable performance across different GNN architectures. Specifically, apart from improving the averaged accuracy, DCGC reduces the cross-architecture standard deviation by an average of 26.3%. In terms of training speed, compared to the current fastest graph condensation methods, DCGC reduces the training time by 96.4%. These results clearly demonstrate the superiority of DCGC in terms of efficacy, generalization ability, and efficiency.

## 2 RELATED WORKS

To reduce the high computational and memory costs when training on large graphs, recent research has started exploring dataset condensation techniques for condensing graph-structured data. As a pioneering work, GCond (Jin et al., 2022) follows the gradient matching paradigm (Zhao et al., 2021) to learn the node features of synthetic nodes. The edges are parameterized as a function of the learned features of two end nodes. SGDD (Yang et al., 2023) follows the similar gradient-matching paradigm in GCond (Jin et al., 2022) but explicitly broadcasts the original graph structure to the synthetic graph to prevent overlooking the structure information in the original graph. SFGC (Zheng et al., 2023) introduces a training trajectory meta-matching scheme for effectively synthesizing small-scale graph-free data. All of the above methods are based on gradient-matching and require specifying a GNN architecture and training its parameters, which not only leads to a challenging training process

but also results in poor cross-architecture generalization ability. To deal with these limitations, this paper explores graph condensation by distribution matching.

We are not the first to utilize distribution matching for graph condensation. The pioneering work GCDM (Liu et al., 2022) treats the original graph and the synthetic graph as two distributions of receptive fields and learns by distribution matching (Zhao & Bilen, 2023). However, GCDM fails to generate the condensed graph of satisfying quality as the gradient-matching methods. Also, GCDM's condensation process still requires specific GNN architectures and complicated bi-level optimization, leaving the limitations of the above gradient-matching methods unresolved.

## 3 METHODOLOGY

### 3.1 PROBLEM FORMULATION AND PRELIMINARIES

**Graph Notations.** Given a graph dataset $\mathcal{G} = \{\mathbf{X}, \mathbf{A}, \mathbf{Y}\}$, where $\mathbf{X} \in \mathbb{R}^{N \times D}$ is the node feature matrix, $\mathbf{A} \in \mathbb{R}^{N \times N}$ is the graph adjacency matrix, and $\mathbf{Y} \in \mathbb{R}^{N \times C}$ is the (one-hot) node label matrix. The target of graph condensation is to learn a synthetic graph $\mathcal{S} = \{\mathbf{X}', \mathbf{A}', \mathbf{Y}'\}$ of $N'$ nodes ($N' \ll N$) such that machine learning models trained on the synthetic dataset $\mathcal{S}$ can perform similarly to those trained on the original graph $\mathcal{G}$. For any class $c = 1, 2 \cdots, C$, we use $N_c$ and $N'_c$ to denote the number of nodes from class $c$ in the original graph $\mathcal{G}$ and synthetic graph $\mathcal{S}$, respectively.

**Graph Condensation via Distribution Matching.** To compress a large dataset into a smaller synthetic one, a natural approach is to ensure that the distribution of synthetic data closely resembles the distribution of real data (Zhao & Bilen, 2023). Denote the original and synthetic graph distribution by $\mathbb{G}$ and $\mathbb{S}$, and the class-conditional ones as $\mathbb{G}_c$ and $\mathbb{S}_c$, respectively. Distribution matching minimizes the discrepancy between $\mathbb{G}$ and $\mathbb{S}$:

$$\min_{\mathcal{S}} \sum_{c=1}^{C} \mathcal{D}(\mathbb{G}_c, \mathbb{S}_c), \quad \text{where } \mathcal{D} \text{ is a measure of distribution difference.} \tag{1}$$

$\mathbb{G}_c(\mathbb{S}_c)$ can be viewed as the subset of the original(synthetic) graph consisting of nodes (and their corresponding contexts) from the $c$-th class. When the distribution of a graph, as well as the difference between the two distributions, is appropriately defined, we are able to solve the graph condensation problem by solving the above optimization problems.

**Maximum Mean Discrepancy.** There are various means to measure the difference between two distributions, and we use Maximum Mean Discrepancy (MMD) since it is differentiable and easy to compute. Given two distributions $\mathbb{G}, \mathbb{S}$, and a mapping $f \in \mathbb{R}^D \to \mathbb{R}$ in the unit ball in a Reproducing Kernel Hilbert Space(RKHS) $\mathcal{H}$, MMD measures the divergence between $\mathbb{G}$ and $\mathbb{S}$ by:

$$\text{MMD}(\mathbb{G}, \mathbb{S}) = \sup_{\|f\|_{\mathcal{H}} \leq 1} (\mathbb{E}_{\mathbb{G}}[f(\boldsymbol{g})] - \mathbb{E}_{\mathbb{S}}[f(\boldsymbol{s})]) = \|\mu_{\mathbb{G}} - \mu_{\mathbb{S}}\|_{\mathcal{H}}, \tag{2}$$

where $\boldsymbol{g} \sim \mathbb{G}, \boldsymbol{s} \sim \mathbb{S}$ are samples, $\mu_{\mathbb{G}} = \mathbb{E}_{\mathbb{G}} \, \phi(\boldsymbol{g}), \mu_{\mathbb{S}} = \mathbb{E}_{\mathbb{S}} \, \phi(\boldsymbol{s})$, and $\phi(\cdot)$ is a feature mapping from $\mathcal{X}$ to $\mathbb{R}$ such that $f(\cdot) = \langle f, \phi(\cdot) \rangle_{\mathcal{H}}$ ($\langle \cdot, \cdot \rangle_{\mathcal{H}}$ denotes the inner product in $\mathcal{H}$). Minimization of $\text{MMD}(\mathbb{G}, \mathbb{S})$ is often reduced to minimization of $\text{MMD}^2(\mathbb{G}, \mathbb{S})$ using the kernel trick:

$$\begin{aligned} \text{MMD}^2(\mathbb{G}, \mathbb{S}) &= \langle \mu_{\mathbb{G}} - \mu_{\mathbb{S}}, \mu_{\mathbb{G}} - \mu_{\mathbb{S}} \rangle_{\mathcal{H}} = \langle \mu_{\mathbb{G}}, \mu_{\mathbb{G}} \rangle_{\mathcal{H}} + \langle \mu_{\mathbb{S}}, \mu_{\mathbb{S}} \rangle_{\mathcal{H}} - 2 \langle \mu_{\mathbb{G}}, \mu_{\mathbb{S}} \rangle_{\mathcal{H}} \\ &= \mathbb{E}_{\mathbb{G}} \langle \phi(\boldsymbol{g}), \phi(\boldsymbol{g}') \rangle_{\mathcal{H}} + \mathbb{E}_{\mathbb{S}} \langle \phi(\boldsymbol{s}), \phi(\boldsymbol{s}') \rangle_{\mathcal{H}} - 2 \, \mathbb{E}_{\mathbb{G}, \mathbb{S}} \langle \phi(\boldsymbol{g}), \phi(\boldsymbol{s}) \rangle_{\mathcal{H}} \\ &= \mathbb{E}_{\mathbb{G}} \kappa(\boldsymbol{g}, \boldsymbol{g}') + \mathbb{E}_{\mathbb{S}} \kappa(\boldsymbol{s}, \boldsymbol{s}') - 2 \, \mathbb{E}_{\mathbb{G}, \mathbb{S}} \kappa(\boldsymbol{g}, \boldsymbol{s}), \end{aligned} \tag{3}$$

where $\kappa(\cdot, \cdot)$ is the kernel function in $\mathcal{H}$, e.g., the Gaussian kernel function $\kappa(\boldsymbol{g}, \boldsymbol{s}) = \exp(-\|\boldsymbol{g} - \boldsymbol{s}\|^2 / (2\sigma^2))$, and $\sigma$ is the bandwidth.

### 3.2 GRAPH DISTRIBUTION AS NODE-WISE DIFFUSION TRAJECTORIES

For i.i.d. data, such as images, the definition of its distribution is self-evident, i.e., the collection of the matrix-form (or vector-form) representations of each single data record, e.g., the pixels of an image (Zhao & Bilen, 2023). However, graph data are non-i.i.d. generated, considering that nodes in the graph are inter-dependent due to the graph structure. While the definition of node feature/label distribution is straightforward, defining the corresponding structural roles of nodes in the entire graph can be challenging. Additionally, we need to fuse node features/labels with their structural attributes to obtain an overall vector form serving as a well-posed representation of their distributions.

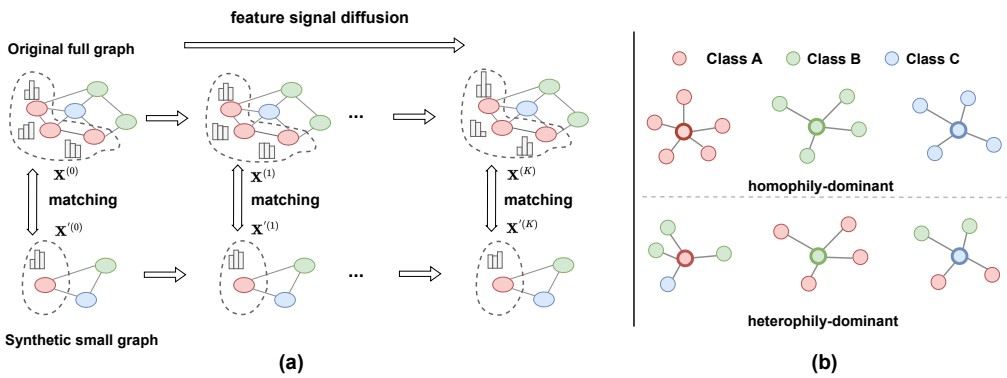

Figure 1: (a) Illustration of graph diffusion process and trajectory matching. The distribution of a graph is defined as the diffusion trajectories of the node features/labels over the graph. The condensed graph is learned by minimizing the MMD distance between the original and synthetic graph. (b) Top: For homophily-dominant graphs, most of the neighboring nodes of the centered node have the same label. Bottom: For heterophily-dominant graphs, connected nodes usually have distinct labels.

**Node feature distribution with graph structure** To resolve this challenge, we resort to graph diffusion process (Kondor & Vert, 2004; Wang et al., 2021; Wu et al., 2023), which utilizes the diffusion ODE to characterize the evolution of a graph signal (e.g., node features) under the spatial constraint of the graph structure:

$$\frac{d\mathbf{X}(t)}{dt} = -\tilde{\mathbf{L}}\mathbf{X}(t), \ \mathbf{X}(0) = \mathbf{X}, \ \tilde{\mathbf{L}} = \mathbf{I} - \tilde{\mathbf{A}}, \ \tilde{\mathbf{A}} = \mathbf{D}^{-1/2}\mathbf{A}\mathbf{D}^{-1/2} \tag{4}$$

In general, Eq. 4 defines the node feature as a type of graph signal, evolving over the graph structure as $t$ increases. At different time points $t$, $\mathbf{X}(t)$ represents the specific response of the node feature signal under the influence of graph structural information to varying degrees, e.g., $\mathbf{X}(0)$ denotes the raw node features without the impact of structure, $\mathbf{X}(1)$ considers the impacts of first-order proximity, whereas $\mathbf{X}(t)$ considers even higher-order graph structure information for large $t$. Since the entire diffusion trajectory is a continuous dynamics that is intractable to compute, we use discretization to obtain a series of $\mathbf{X}(t)$ in discrete time steps. With the explicit Euler's method, we have

$$\mathbf{X}(t + \Delta t) \approx \mathbf{X}(t) - \Delta t \mathbf{L}\mathbf{X}(t) = [(1 - \Delta t)\mathbf{I} + \Delta t\tilde{\mathbf{A}}]\mathbf{X}(t). \tag{5}$$

The explicit diffusion scheme in Eq. 5 is determined by the time interval $\Delta t$, and it is proved to be stable under the following conditions:

**Proposition 1.** *(Numerical stability, Theorem 1 in Chamberlain et al. (2021)) The step-wise diffusion scheme in Eq. 5 is stable for $0 < \Delta t \le 1$.*

The proof is in Appendix B. Therefore, for a given time interval $\Delta t$ and step $k$, we are able to obtain the signals of nodes at step $k$, in the following format:

$$\mathbf{X}^{(k)} = \mathbf{X}(k \cdot \Delta t) = [(1 - \Delta t)\mathbf{I} + \Delta t\tilde{\mathbf{A}}]^k \mathbf{X}^{(0)}. \tag{6}$$

With steps $k = 0, 1, 2, ..., K$, we can obtain a sequence of graph feature signals influenced by increasing degrees of the graph structure. Based on this, we define the distribution of the node features over a given graph structure as the trajectories of the diffusion process with finite $K$ steps:

**Definition 1.** *Given a graph $\mathcal{G}$ with initial node features $\mathbf{X}^{(0)} = \mathbf{X}$, the adjacency matrix $\mathbf{A}$, and the maximum step $K$. We define the **node feature distribution** over $\mathcal{G}$ at step $k$, $\mathbb{G}^{f,(k)}$, as follows:*

$$\mathbb{G}^{f,(k)} \triangleq \mathbf{X}^{(k)}, \ \boldsymbol{g}_i^{f,(k)} = \boldsymbol{x}_i^{(k)} \sim \mathbb{G}^{f,(k)}, \ k = 0, 1, \cdots, K. \tag{7}$$

*where $\boldsymbol{g}_i^{f,(k)}$ denotes the feature signal of node $i$ at step $k$, and the superscript $^f$ is short for $\underline{f}$eature.*

For the class-conditional version, we use the subscript $c$ to denote the corresponding symbols specified to nodes from class $c$, i.e., $\mathbb{G}_c^{f,(k)} \triangleq \mathbf{X}_c^{(k)}, \boldsymbol{g}_{c,i}^{f,(k)} = \boldsymbol{x}_{c,i}^{(k)} \sim \mathbb{G}_c^{f,(k)}$.

**Node label distribution with graph structure**  In addition to the distribution of node features over the graph structure, we further extend the definition to the distribution of node labels in the graph. Our approach is motivated by an observation that real-world graphs usually exhibit different interconnecting patterns for nodes belonging to different classes, e.g., homophily v.s. heterophily. We provide an illustrative example in Fig. 1 (b): In homophilic graphs, a node is usually connected to other nodes sharing the same label. Therefore, we expect that nodes from the same class are also connected in the condensed graph. By contrast, when the original graph is a heterophilic one, e.g., nodes from different classes are more likely to connect, we wish the condensed graph had similar properties. In this regard, we consider a similar diffusion process to the one-hot node label matrix $\mathbf{Y}$ and obtain the signal of the node label matrix at step $k$ as:

$$\mathbf{Y}^{(k)} = \ell_1 \text{norm} \left[ \left( (1 - \Delta t)\mathbf{I} + \Delta t \tilde{\mathbf{A}} \right)^k \mathbf{Y} \right] \in \mathbb{R}^{N \times C}. \tag{8}$$

Notice that here we apply $\ell_1$ normalization such that the entry $\mathbf{Y}_{i,c}^{(k)}$ represents the proportion of label $c$ in node $i$'s $k$-hop neighborhoods. Then, we define the label distribution over a graph as the labels' evolution in a diffusion process:

**Definition 2.** *Given a graph $\mathcal{G}$ with adjacency matrix $\mathbf{A}$ and node label matrix $\mathbf{Y}$, and the maximum step $K$. We define its **label distribution over graph** at step $k$ as follows:*

$$\mathbb{G}^{l,(k)} \triangleq \mathbf{Y}^{(k)}, \ \boldsymbol{g}_i^{l,(k)} = \boldsymbol{y}_i^{(k)} \sim \mathbb{G}^{l,(k)}, \ k = 0, \cdots, K \tag{9}$$

*where $\boldsymbol{g}_i^{l,(k)}$ denotes the label signal of node $i$ at step $k$, and the superscript $l$ is short for label.*

Similarly, we use $\mathbb{G}_c^{l,(k)}$ and $\boldsymbol{g}_{c,i}^{l,(k)}$ to denote the corresponding class-conditional versions. Note that the above notations are for the original graph $\mathcal{G}$ and its distribution $\boldsymbol{g} \sim \mathbb{G}$. For the synthetic graph $\mathcal{S}$, we use $\boldsymbol{s} \sim \mathbb{S}$ instead.

## 3.3 Training of DCGC

The training of our model DCGC seeks to learn the synthetic graph by minimizing the MMD between the feature/label distribution of the original full graph $\mathcal{G}$ and the synthetic graph $\mathcal{S}$ of class $c$ at step $k$:

$$\mathcal{L}_c^{f,(k)} = \text{MMD}^2(\mathbb{G}_c^{f,(k)}, \mathbb{S}_c^{f,(k)}) = \mathbb{E}_{\mathbb{G}} \kappa(\boldsymbol{g}_c^{f,(k)}, \boldsymbol{g}_c^{'f,(k)}) + \mathbb{E}_{\mathbb{S}} \kappa(\boldsymbol{s}_c^{f,(k)}, \boldsymbol{s}_c^{'f,(k)}) - 2 \mathbb{E}_{\mathbb{G},\mathbb{S}} \kappa(\boldsymbol{g}_c^{f,(k)}, \boldsymbol{s}_c^{f,(k)})$$

$$= \sum_{i=1}^{N_c'} \sum_{j=1}^{N_c'} \kappa(\boldsymbol{s}_{c,i}^{f,(k)}, \boldsymbol{s}_{c,j}^{f,(k)}) - 2 \sum_{i=1}^{N_c} \sum_{j=1}^{N_c'} \kappa(\boldsymbol{g}_{c,i}^{f,(k)}, \boldsymbol{s}_{c,j}^{f,(k)}). \tag{10}$$

The last step discards the term $\mathbb{E}_{\mathbb{G}} \kappa(\boldsymbol{g}_c^{f,(k)}, \boldsymbol{g}_c^{'f,(k)})$ since it only depends on the original graph $\mathcal{G}$ and is not involved in the optimization process. Similarly, the label distribution matching loss of class $c$ at step $k$ can be derived as:

$$\mathcal{L}_c^{l,(k)} = \sum_{i=1}^{N_c'} \sum_{j=1}^{N_c'} \kappa(\boldsymbol{s}_{c,i}^{l,(k)}, \boldsymbol{s}_{c,j}^{l,(k)}) - 2 \sum_{i=1}^{N_c} \sum_{j=1}^{N_c'} \kappa(\boldsymbol{g}_{c,i}^{l,(k)}, \boldsymbol{s}_{c,j}^{l,(k)}). \tag{11}$$

**Contrastive Alignment**  Minimizing Eq. 10 and Eq. 11 enables the condensed graph to preserve the $k$-th order feature/label distribution over the graph structure. Yet, it does not ensure the alignment between the node features and labels in the condensed graph. For example, given the feature distribution $\{\boldsymbol{s}_1^f, \boldsymbol{s}_2^f, \cdots, \boldsymbol{s}_{C'}^f\}$ and label distribution $\{\boldsymbol{s}_1^l, \boldsymbol{s}_2^l, \cdots, \boldsymbol{s}_{C'}^l\}$ of the synthetic graph $\mathcal{S}$, the exchange of entries (e.g., $\boldsymbol{s}_1^f \leftrightarrow \boldsymbol{s}_2^f, \boldsymbol{s}_3^l \leftrightarrow \boldsymbol{s}_4^l$) does not affect the values of losses. Yet, $\boldsymbol{s}_1^f$ and $\boldsymbol{s}_1^l$ no longer correspond to the signals of the same node, i.e., node 1 in the synthetic graph.

To address the above issue, we leverage the contrastive InfoNCE loss (van den Oord et al., 2018) for aligning the node-wise feature distribution and label distribution of the condensed graph:

$$\mathcal{L}_c^{\text{reg},(k)} = -\sum_{i=1}^{N_c'} \log \frac{\exp(\phi(\boldsymbol{s}_{c,i}^{f,(k)})^\top \cdot \psi(\boldsymbol{s}_{c,i}^{l,(k)})/\tau)}{\sum\limits_{j=1}^{N_c'} \exp(\phi(\boldsymbol{s}_{c,i}^{f,(k)})^\top \cdot \psi(\boldsymbol{s}_{c,j}^{l,(k)})/\tau)}, \tag{12}$$

where $\phi(\cdot) : \mathbb{R}^D \to \mathbb{R}^d$ and $\psi(\cdot) : \mathbb{R}^D \to \mathbb{R}^d$ are two linear projectors mapping $s^f$ and $s^l$ to the same dimension and $\tau$ is the temperature hyper-parameter. We set $d = D$ and $\tau = 0.5$ in this work. Based on contrastive learning, Eq. 12 maximizes the mutual information between the feature signal $s_i^f$ and label signal $s_i^l$ of the same node $i$, encouraging their alignment.

**Overall learning objective**  Integrating Eq. 10, Eq. 11, and Eq. 12 over all classes and time steps, we obtain the overall learning objective:

$$\min_{\mathbf{X}',\mathbf{A}',\phi,\psi} \mathcal{L} = \sum_{k=0}^{K} \sum_{c=1}^{C} \left( \mathcal{L}_c^{f,(k)} + \mathcal{L}_c^{l,(k)} + \mathcal{L}_c^{\text{reg},(k)} \right). \qquad (13)$$

We provide an algorithmic illustration of the condensation process in Algorithm 1 in Appendix A.

**Complexity**  Finally, we analyze the complexity of DCGC. 1) Computing the feature/label distribution of the original graph $\mathcal{G}$ requires $\mathcal{O}(EK(D + C))$, while this process is non-parametric and can be obtained via one-step preprocessing. Therefore, the computation overhead of this step is negligible compared with the entire condensation process. 2) Computing the feature/label distribution of the condensed graph requires $\mathcal{O}(E'K(D + C)) = \mathcal{O}(N'^2K(D + C))$. 3) Computing the MMD loss for all $k$ and $c$ takes $\mathcal{O}(K \cdot (D + C) \cdot \sum_{c=1}^{C} N_c'(N_c + N_c'))$, which depends on the number of nodes in each class. Yet, notice that $\sum_{c=1}^{C} N_c'(N_c + N_c') \leq N'(N + N')$, and the equality holds if and only if there is only one class. 4) Similarly, the contrastive alignment loss takes $\mathcal{O}(KDN'^2)$. Combining all the steps together, the overall complexity of DCGC is $\mathcal{O}(K(D + C)NN')$. Considering that $N' \ll N$ and $K$ is small in practice, the overall complexity is slightly greater than $\mathcal{O}(N)$ and much smaller than $\mathcal{O}(N^2)$, and therefore DCGC is time and memory-efficient.

## 4    EXPERIMENTS

### 4.1    EXPERIMENTAL SETUPS

**Datasets.** Following previous literature (Jin et al., 2022; Liu et al., 2022), we mainly evaluate the quality of the condensed graphs on six node classification datasets: Cora, Citeseer, Pubmed (Yang et al., 2016), Flickr, Reddit (Zeng et al., 2020), and Ogbn-arXiv (Hu et al., 2020). For a fair comparison, we use the public splits for all datasets. We also The details of these datasets are deferred to Appendix C.1. In addition to the above-mentioned homophily-dominated graphs, we consider two more heterophilic graphs, Chameleon and Actor.

**Competitors.** We compare our proposed method with four SOTA graph condensation methods: GCond (Jin et al., 2022), GCDM (Liu et al., 2022), SGDD (Yang et al., 2023), and SFGC (Zheng et al., 2023). Following (Jin et al., 2022), we also compare with three traditional selection-based methods: Herding (Welling, 2009), K-center (Sener & Savarese, 2018), and graph coarsening (Huang et al., 2021). The training performance using the original full graph is provided for reference as well.

**Implementation Details.** We implement the proposed method with Pytorch and DGL (Wang et al., 2019). In the training stage, we first initialize the node feature matrix $\mathbf{X}'$ and $\mathbf{A}'$ according to the proposed strategies. Then $\mathbf{X}'$ and $\mathbf{A}'$ are optimized using Eq. 13. In the evaluation stage, we train a 2-layer GCN model (Kipf & Welling, 2017) of hidden dimension 512 using the condensed graph and then report the accuracy on the testing nodes of the original graph. We repeat all experiments for 20 times and report the average performance with standard deviation. We provide more implementation details in Appendix C.3.

Given a condensation ratio $r$, the number of nodes in the condensed graph is $N' = N \times r$. Then, we initialize the labels of a condensed graph $\mathbf{Y}'$ such that the proportion of each class in the condensed graph is the same as that in the original full graph, i.e., $\frac{N_c'}{N'} = [\frac{N_c}{N}]$, and $\sum_c N_c' = N$. The optimization of Eq. 13 is straightforward, yet might be challenging if the initial states of parameters ($\mathbf{X}'$ and $\mathbf{A}'$) are far from the optimal ones. Empirically, we found that the traditional random initialization methods (e.g., Xavier initialization) lead to poor performance due to the difficulty in optimization. To this end, we adopt a simple strategy to initialize the node feature matrix $\mathbf{X}'$ and the graph adjacency matrix $\mathbf{A}'$. For each class $c$, we randomly select $N_c'$ nodes from the original graph having the same

Table 1: Comparison with SOTA methods regarding testing accuracy (%). **Bold entries are the best results**. DCGC outperforms existing methods on almost all datasets and all condemnation ratios.

| Dataset | Ratio ($r$) | Other graph size reduction methods | | | Condensation Methods | | | | | Whole |
|---|---|---|---|---|---|---|---|---|---|---|
| | | Herding | K-Center | Coarsening | GCond | GCDM | SGDD | SFGC | DCGC | |
| Cora | 1.30% | 67.0±1.3 | 64.0±2.3 | 31.2±0.2 | 79.8±1.3 | 69.4±1.3 | 80.1±0.7 | 80.1±0. | **81.1±0.6** | 82.7±0.5 |
| | 2.60% | 73.4±1.0 | 73.2±1.2 | 65.2±0.6 | 80.1±0.6 | 77.2±0.4 | 80.6±0.8 | 81.9±0.5 | **81.7±0.6** | |
| | 5.20% | 76.8±0.1 | 76.7±0.1 | 70.6±0.1 | 79.3±0.3 | 79.4±0.1 | 80.4±1.6 | 81.6±0.8 | **82.1±0.5** | |
| Citeseer | 0.90% | 57.1±1.5 | 52.4±2.8 | 52.2±0.4 | 70.5±1.2 | 62.0±0.1 | 69.5±0.4 | 71.4±0.5 | **71.6±0.6** | 72.4±0.4 |
| | 1.80% | 66.7±1.0 | 64.3±1.0 | 59.0±0.5 | 70.6±0.4 | 69.5±1.1 | 70.2±0.8 | **72.4±0.4** | 72.2±0.5 | |
| | 3.60% | 69.0±0.1 | 69.1±0.1 | 65.3±0.5 | 69.8±1.4 | 69.8±0.2 | 70.3±1.7 | 70.6±0.7 | **72.7±0.5** | |
| Pubmed | 0.08% | 76.7±0.7 | 64.5±2.7 | 18.1±0.1 | 76.5±0.2 | 75.7±0.3 | 76.7±0.4 | 77.1±0.5 | **78.4±0.5** | 79.8±0.4 |
| | 0.15% | 76.2±0.5 | 69.4±0.7 | 28.7±4.1 | 77.1±0.5 | 77.3±0.1 | 77.5±0.4 | 77.6±0.5 | **78.9±0.3** | |
| | 0.30% | 78.0±0.5 | 69.1±0.1 | 65.3±0.5 | 77.9±1.4 | 78.3±0.9 | 78.2±0.8 | 78.8±0.6 | **79.5±0.3** | |
| Flickr | 0.10% | 42.5±1.8 | 42.0±0.7 | 41.9±0.2 | 46.5±0.4 | 46.8±0.2 | 46.9±0.1 | 46.6±0.6 | **47.6±0.3** | 50.2±0.3 |
| | 0.50% | 43.9±0.9 | 43.2±0.1 | 44.5±0.1 | 47.1±0.1 | 47.9±0.3 | 47.1±0.3 | 47.0±0.1 | **48.2±0.3** | |
| | 1.00% | 44.4±0.6 | 44.1±0.4 | 44.6±0.1 | 47.1±0.1 | 47.5±0.1 | 47.1±0.1 | 47.1±0.1 | **48.9±0.1** | |
| Reddit | 0.05% | 53.1±2.5 | 46.6±2.3 | 40.9±0.5 | 88.0±1.8 | 86.5±1.1 | 90.5±2.1 | 89.7±0.2 | **90.8±1.4** | 93.9±0.0 |
| | 0.10% | 62.7±1.0 | 53.0±3.3 | 42.8±0.8 | 89.6±0.7 | 88.3±0.8 | **91.6±1.0** | 90.0±0.3 | 91.5±0.9 | |
| | 0.20% | 71.0±1.6 | 58.5±2.1 | 47.4±0.9 | 90.1±0.5 | 89.2±0.7 | 91.6±1.8 | 89.9±0.4 | **92.0±0.6** | |
| arXiv | 0.05% | 52.4±1.8 | 47.2±3.0 | 35.4±0.3 | 59.2±1.1 | 56.2±0.3 | 60.8±1.3 | **65.5±0.7** | 65.1±0.7 | 71.4±0.1 |
| | 0.25% | 58.6±1.2 | 56.8±0.8 | 43.5±0.2 | 63.2±0.3 | 59.6±0.4 | 65.8±1.2 | 66.1±0.4 | **66.8±0.3** | |
| | 0.50% | 60.4±0.8 | 60.3±0.4 | 50.4±0.1 | 64.0±0.4 | 62.4±0.1 | 66.3±0.8 | 66.8±0.4 | **67.9±0.4** | |

label and use their features to initialize $\mathbf{X}'_c$. In this way, we wish the synthetic graph had individual node features similar to those of the original graph.

For the adjacency matrix $\mathbf{A}'$, we initialize its on-diagonal terms to be a value $\epsilon_{\mathsf{on}}$ close to 1, while off-diagonal terms to be a small value $\epsilon_{\mathsf{off}}$. In this way, we initialize a synthetic graph that primarily consists of self-loops, thereby reducing the noisy edges that random initialization may introduce. The parameters of $\mathbf{A}'$ are obtained via the Sigmoid function such that they are restricted within the range $(0, 1)$. Note that the obtained condensed graph $\mathcal{S}$ with adjacency matrix $\mathbf{A}'$ is a dense, undirected graph with edge weights in the range $(0, 1)$.

**Hyperparameter settings.** For the initialization of the adjacency matrix $\mathbf{A}'$, we set $\varepsilon_{\mathsf{on}} = 0.999$. and $\varepsilon_{\mathsf{off}} = 0.001$. The diffusion time interval is set as $\Delta t = 1$, and the maximum diffusion step is set as $K = 3$. For the bandwidth of the Gaussian kernel function when computing the MMD distance, we set $2\sigma^2$ as the median $\ell_2$ distance of the samples since it is dataset-sensitive.

## 4.2    MAIN RESULTS

**Comparison on common benchmarks.** In Table 1, we present the performance comparison between the proposed DCGC and the baseline methods under node classification tasks. The experimental results demonstrate that our proposed method performs on par or even better than SOTA gradient-matching methods on all datasets and condensation ratios, which strongly illustrates the effectiveness of DCGC across different datasets.

**Cross-architecture generalization performance.** One important limitation of existing methods is that they all rely on a predefined GNN encoder during the condensation process, which might lead to poor cross-architecture generalization ability. In this section, we empirically validate the generalization ability of the proposed DCGC on Cora, Citeseer, Pubmed, and Ogbn-arXiv. The condensation process of DCGC involves no encoders. In evaluation, we consider different-architectured GNN classifiers: GCN (Kipf & Welling, 2017), GraphSAGE (Hamilton et al., 2017), GAT (Velickovic et al., 2018), Cheby (Defferrard et al., 2016), and APPNP (Klicpera et al., 2019). We also report the average performance with standard deviation across different architectures. A small standard deviation indicates that the condensed graph has relatively stable performance across classifiers with different architectures, so a model with a higher average accuracy and a smaller standard deviation is preferred. As demonstrated in Table 2, the proposed DCGC achieves the highest average accuracy across different GNN architectures on the four datasets. In addition, DCGC achieves the lowest Std., indicating its superior generalization ability across different architectures. These results clearly demonstrate the superior advantages of DCGC as a data-centric condensation method.

Table 2: Cross-architecture generalization performance comparison. The condensed graphs are obtained via GCN (except DCGC, which is data-centric), while tested using six different GNN architectures: SGC, GCN, SAGE, GAT, Cheby, and APPNP, and the overall performance is reflected by the average testing accuracy (Avg.) and its standard deviation (Std.).

| Datasets | Methods | Architectures | | | | | | Statistics | |
|---|---|---|---|---|---|---|---|---|---|
| | | SGC | GCN | SAGE | GAT | Cheby | APPNP | Avg. | Std. |
| **Cora** $r = 2.6\%$ | GCond | 79.3 | 80.1 | 78.2 | 66.2 | 76.0 | 78.5 | 76.4 | 5.18 |
| | GCDM | 78.7 | 79.4 | 78.5 | 73.2 | 75.4 | 77.8 | 77.2 | 2.38 |
| | SGDD | 78.5 | 79.8 | 80.4 | 75.8 | 78.5 | 78.4 | 78.6 | 1.59 |
| | SFGC | 79.1 | 81.1 | 81.9 | 80.8 | 79.0 | 78.8 | 80.1 | 1.31 |
| | DCGC | 80.7 | 81.7 | 81.5 | 82.1 | 80.6 | 82.3 | **81.4** | **0.80** |
| **Citeseer** $r = 1.8\%$ | GCond | 70.3 | 70.6 | 66.2 | 55.4 | 68.3 | 69.6 | 66.7 | 5.78 |
| | GCDM | 68.7 | 69.5 | 67.1 | 62.5 | 68.9 | 69.1 | 67.6 | 2.65 |
| | SGDD | 69.9 | 70.2 | 67.8 | 65.7 | 68.5 | 70.7 | 68.8 | 1.87 |
| | SFGC | 71.8 | 71.6 | 71.7 | 72.1 | 71.8 | 70.5 | 71.6 | 0.56 |
| | DCGC | 70.6 | 72.2 | 70.5 | 71.2 | 69.4 | 72.0 | **71.8** | **0.44** |
| **Pubmed** $r = 0.15\%$ | GCond | 75.8 | 77.1 | 76.2 | 74.8 | 73.5 | 77.9 | 75.9 | 1.58 |
| | GCDM | 76.5 | 77.3 | 75.7 | 77.9 | 75.4 | 78.2 | 76.8 | 1.15 |
| | SGDD | 77.1 | 77.5 | 76.9 | 76.8 | 76.2 | 78.7 | 77.2 | 0.85 |
| | SFGC | 76.8 | 77.6 | 77.4 | 77.1 | 75.3 | 78.2 | 77.2 | 0.69 |
| | DCGC | 78.6 | 78.9 | 78.6 | 79.4 | 78.4 | 79.5 | **78.9** | **0.46** |
| **Ogbn-arXiv** $r = 0.25\%$ | GCond | 63.7 | 63.2 | 62.6 | 60.0 | 54.9 | 63.4 | 61.3 | 3.41 |
| | GCDM | 61.9 | 61.6 | 55.4 | 61.7 | 44.5 | 52.3 | 56.2 | 6.99 |
| | SGDD | 59.8 | 64.2 | 61.2 | 62.7 | 53.7 | 60.1 | 60.3 | 3.62 |
| | SFGC | 64.8 | 65.1 | 64.8 | 65.7 | 60.7 | 63.9 | 64.2 | 1.79 |
| | DCGC | 64.6 | 66.8 | 65.2 | 65.9 | 62.3 | 65.8 | **65.1** | **1.56** |

Table 3: Performance of graph condensation methods on heterophilic graphs. The column **C** denotes the GNN model for condensation, while the row **T** denotes the GNN model for evaluation (training/testing). GNNs good at heterophilic graphs fail on graphs condensed by homophilic GNNs.

| Datasets | C \ T | GCN | | | | GPRGNN | | | |
|---|---|---|---|---|---|---|---|---|---|
| | | GCond | SFGC | DCGC | Full | GCond | SFGC | DCGC | Full |
| **Amazon-Rating** $r = 0.4\%$ | GCN | 44.37 | 46.43 | **46.98** | 48.70 | 41.18 | 42.06 | **43.71** | 44.88 |
| | GPRGNN | 40.92 | 42.38 | | | 41.59 | 42.89 | | |
| **Actor** $r = 1.3\%$ | GCN | 27.35 | 28.92 | **29.93** | 30.59 | 31.29 | 34.11 | **38.77** | 39.30 |
| | GPRGNN | 28.41 | 29.18 | | | 36.68 | 38.48 | | |

**Performance on condensing heterophilic graphs** Another potential limitation of existing gradient-matching methods is that when the model used for graph condensation is sub-optimal, the subsequent GNN model might suffer from significant performance degradation even if it has a proper architecture. To verify this, we perform experiments on two heterophilic datasets, Chameleon and Actor (Pei et al., 2020). In Table 3, we present the performance of GCond and SFGC using distinct GNN architectures for condensing heterophilic graphs. Note that GCN (Kipf & Welling, 2017) usually performs sub-optimally on heterophilic graphs, while GPR-GNN (Chien et al., 2021) is good at both homophilic and heterophilic graphs. We can observe that when using the same architecture for condensation and testing, all methods achieve close performance to training the architecture on the full graph. However, graphs condensed by GCN fail to give satisfying performance when evaluated using GPRGNN and are far from training GPRGNN on the full graph. By contrast, the proposed DCGC is able to give a consistently close performance to the full graph regardless of the architecture.

### 4.3 ABLATION STUDIES AND EFFICIENCY COMPARISON

**Effects of the components in** DCGC**.** Next, we investigate the importance of each component of DCGC. The loss function of DCGC (in Eq. 13) consists of three parts: feature-level matching

Table 4: Performance of removing feature/label/reg loss on Ogbn-arXiv dataset.

| Variants | $r = 0.05\%$ | $r = 0.25\%$ | $r = 0.50\%$ |
|---|---|---|---|
| w/o $\mathcal{L}^f$ | 56.1 13.8%↓ | 59.8 10.5%↓ | 61.1 10.0%↓ |
| w/o $\mathcal{L}^l$ | 34.3 47.3%↓ | 39.9 40.3%↓ | 43.2 36.4%↓ |
| w/o $\mathcal{L}^{reg}$ | 63.9 2.0%↓ | 66.2 0.9%↓ | 66.9 1.5%↓ |
| DCGC | **65.1** | **66.8** | **67.9** |

Table 5: Training time on Ogbn-arXiv dataset for 50 epochs, on an Nvidia 4090.

| $r$ | GCond | GCDM | SGDD | DCGC |
|---|---|---|---|---|
| 0.05% | 351 s | 325 s | 349 s | **11.69 s** |
| 0.25% | 448 s | 358 s | 417 s | **12.21 s** |
| 0.50% | 603 s | 411 s | 576 s | **13.84 s** |

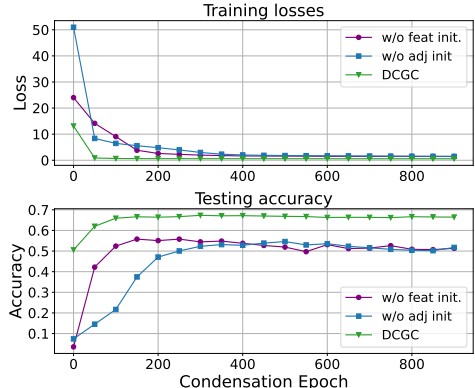

Figure 2: Ablation study on initialization strategies

loss, label-level matching loss, and alignment loss, while the last only takes effect when both the former ones exist. Therefore, we investigate the impact of using each individual loss separately on the performance of DCGC. Note that removing either $\mathcal{L}^f$ or $\mathcal{L}^l$ indicates that $\mathcal{L}^{reg}$ is removed as well. In Table 4, we present the results on Ogbn-arXiv dataset. It is observed that using merely the feature-level matching loss can only achieve sub-optimal performance. This indicates that solely considering the feature distribution over the graph is insufficient to capture the distribution of the entire graph, especially when the graph's structure is complex and there are a significant number of labeled nodes. Furthermore, merely using the label-level matching loss results in extremely poor performance (an average accuracy drop of 40%), which underscores the importance of node features.

**Effects of the** DCGC**'s initialization strategies.** Next, we investigate the importance of the initialization strategies, which are assessed by removing the feature matrix initialization and adjacency matrix initialization from DCGC, respectively. In Fig. 2, we present the training curves of training loss and test accuracy w.r.t. the epoch on Ogbn-arXiv dataset ($r = 0.5\%$). It can be observed that with the proposed two initialization strategies, the initial loss is set to be very low, resulting in a good starting point in the optimization space. This not only significantly accelerates the model's convergence speed but also makes it easier for the model to converge to better values, reducing the risk of getting stuck in local optima. Removing any one of the initialization methods significantly increases the training difficulty of the model, which may lead to sub-optimal performance.

**Comparison of training time.** Finally, we validate the efficiency of the proposed DCGC by comparing its training time with SOTA graph condensation methods. Following previous evaluation settings (Jin et al., 2022; Yang et al., 2023), we report the training time of 50 epochs on Ogbn-arXiv dataset in Table 5. As shown in Table 5, DCGC achieves a much faster training speed compared with existing methods for all condensation ratios. To be specific, DCGC reduces the epoch-wise training time by 96.4% Furthermore, as the graph condensation $r$ increases, the training time of DCGC increases to a lesser extent compared to other methods. This indicates that our proposed DCGC exhibits better scalability relative to other methods.

## 5 CONCLUSIONS

In this paper, we have proposed DCGC for condensing a large-scale graph into a small one. We define the distribution of a graph as the trajectories obtained by conducting diffusion on its nodes' features (as well as labels) over the graph structure. Following the idea of distribution matching, we learn a small-scale graph by minimizing the Maximum Mean Discrepancy (MMD) distance between the distribution of the original graph and the synthetic graph. To address the convergence challenges in distribution matching, we propose a sophisticated parameter initialization strategy that not only accelerates convergence but also reduces the risk of getting stuck in local optima. Extensive experimental results demonstrate that our proposed DCGC achieves state-of-the-art results on the selected datasets and exhibits excellent cross-architecture generalization ability. Moreover, it significantly reduces the training time for condensation compared to current methods.

## REPRODUCIBILITY STATEMENT

We provide the algorithms of DCGC in Appendix A, the PyTorch-style pseudo codes for implementing DCGC in Appendix C.3, and the proof in Appendix B. The detailed hyperparameter settings are in Section 4.1.

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

## A ALGORITHMS

We provide the algorithmic illustration of the condensation of DCGC in Algorithm 1.

---

**Algorithm 1:** Algorithm for DCGC

---

**Input:** A graph $\mathcal{G} = (\mathbf{X}, \mathbf{A}, \mathbf{Y}) = (\mathcal{V}, \mathcal{E})$ with $N$ nodes and $E$ edges, where $\mathbf{X} \in \mathbb{R}^{N \times D}$ is node feature matrix, $\mathbf{A} \in \mathbb{R}^{N \times N}$ is the adjacency matrix, and $\mathbf{Y} \in \mathbb{R}^{N \times C}$ is the one-hot node label matrix. Condensation ratio $r$. Diffusion steps $K$. Kernel function: $\kappa(\boldsymbol{x}, \boldsymbol{y}) = \exp(-\|\boldsymbol{x} - \boldsymbol{y}\|^2/(2\sigma^2))$.

**Output:** A condensed graph $\mathcal{S} = (\mathbf{X}', \mathbf{A}', \mathbf{Y}')$. $\mathbf{X}' \in \mathbb{R}^{N' \times D}, \mathbf{A}' \in \mathbb{R}^{N' \times N'}, \mathbf{Y}' \in \mathbb{R}^{N' \times C}$

1   Number of Labels: $N' = [N \times r]$
2   Generate $\mathbf{Y}'$
3   Initialize $\mathbf{X}'$ and $\mathbf{A}'$
4   **for** $k \in [0, K]$ **do**
5      $\mathbf{X}^{(k)} = \tilde{\mathbf{A}}^k \mathbf{X} \in \mathbb{R}^{N \times D}$
6      $\mathbf{Y}^{(k)} = \tilde{\mathbf{A}}^k \mathbf{Y}/\|\tilde{\mathbf{A}}^k \mathbf{Y}\|_1 \in \mathbb{R}^{N \times C}$
7   **while** *not converging* **do**
8      **for** $k \in [0, K]$ **do**
9          $\mathbf{X}'^{(k)} = \tilde{\mathbf{A}}'^k \mathbf{X}' \in \mathbb{R}^{N' \times D}$
10         $\mathbf{Y}'^{(k)} = \tilde{\mathbf{A}}'^k \mathbf{Y}'/\|\tilde{\mathbf{A}}'^k \mathbf{Y}'\| \in \mathbb{R}^{N' \times C}$
11      **for** $k \in [0, K]$ **do**
12         **for** $c \in [1, C]$ **do**

13            
$$\mathcal{L}_c^{f,(k)} = \left( \sum_{i=1}^{N_c'} \sum_{j=1}^{N_c'} \kappa(\boldsymbol{s}_{c,i}^{f,(k)}, \boldsymbol{s}_{c,j}^{f,(k)}) \right) - 2 \sum_{c=1}^{C} \left( \sum_{i=1}^{N_c} \sum_{j=1}^{N_c'} \kappa(\boldsymbol{g}_{c,i}^{f,(k)}, \boldsymbol{s}_{c,j}^{f,(k)}) \right)$$

14            
$$\mathcal{L}_c^{l,(k)} = \left( \sum_{i=1}^{N_c'} \sum_{j=1}^{N_c'} \kappa(\boldsymbol{s}_{c,i}^{l,(k)}, \boldsymbol{s}_{c,j}^{l,(k)}) \right) - 2 \sum_{c=1}^{C} \left( \sum_{i=1}^{N_c} \sum_{j=1}^{N_c'} \kappa(\boldsymbol{g}_{c,i}^{l,(k)}, \boldsymbol{s}_{c,j}^{l,(k)}) \right)$$

15            
$$\mathcal{L}_c^{\text{reg},(k)} = -\sum_{i=1}^{N_c'} \log \frac{\exp(\phi(\boldsymbol{s}_{c,i}^{f,(k)})^\top \cdot \psi(\boldsymbol{s}_{c,i}^{l,(k)})/\tau)}{\sum_{j=1}^{N_c'} \exp(\phi(\boldsymbol{s}_{c,i}^{f,(k)})^\top \cdot \psi(\boldsymbol{s}_{c,j}^{l,(k)})/\tau)}$$

16      $\mathcal{L} = \sum_{k=0}^{K} \sum_{c=1}^{C} \left( \mathcal{L}_c^{f,(k)} + \mathcal{L}_c^{l,(k)} + \mathcal{L}_c^{\text{reg},(k)} \right)$
17      Gradient backward to update $\mathbf{X}', \mathbf{A}', \phi, \psi$

---

## B PROOF

### B.1 PROOF FOR PROPOSITION 1

*Proof.* First, we can rewrite Eq. 5 as follows:

$$\mathbf{X}^{(t+\Delta t)} = ((1 - \Delta t)\mathbf{I} + \Delta t \tilde{\mathbf{A}})\mathbf{X}^{(t)} = \mathbf{Q}\mathbf{X}^{(t)}$$

where $\mathbf{Q} = (1 - \Delta t)\mathbf{I} + \Delta t \tilde{\mathbf{A}}$.

We require that the amplification factor $\|\mathbf{Q}\| \leq 1$. It is sufficient to show that $\mathbf{Q}$ is a right stochastic matrix, which has the property that its spectral radius $\lambda_{max} \leq 1$. Q is right stochastic if:

1. $\sum_{j=1}^{N} q_{ij} = 1$.

2. $q_{ij} > 0, \forall i, j$

Condition 1 is met since $\sum_{j=1}^{N} q_{ij} = (1 - \Delta t) + \Delta t \sum_{j=1}^{N} \tilde{a}_{ij} = 1$.

When $i \neq j$, Condition 2 is met, since $q_{ij} = \Delta t \cdot \tilde{a}_{ij} > 0$. When $i = j$, we require $q_{ii} = 1 - \Delta t + \Delta t \cdot \tilde{a}_{ii} = 1 - (1 - \tilde{a}_{ii})\Delta t > 0$, which indicates $\Delta t < \frac{1}{1 - \tilde{a}_{ii}} \Leftrightarrow 0 < \Delta t \leq 1$.

Therefore, the proof is complete. $\square$

## C EXPERIMENTAL DETAILS

### C.1 DATASETS

We evaluate the proposed methods on three widely used small-scale citation networks: Cora, Citeseer, and Pubmed, two large-scale graphs, Flickr and Reddit from the GraphSAINT (Zeng et al., 2020) paper, one large-scale graph Ogbn-arXiv from Open Graph Benchmark (OGB) (Hu et al., 2020), and two heterophilic graphs obtained from Geom-GCN paper (Pei et al., 2020). In Table 6, we provide the statistical information of these datasets, including the number of nodes, number of classes, number of classes, the feature dimension, and the training/validation/testing split of the original graph.

Table 6: Statistics of datasets

| Datasets | #Nodes | #Edges | #Classes | #Features | Training / Validation / Testing |
|---|---|---|---|---|---|
| Cora | 2,708 | 10,556 | 7 | 1,433 | 140/1500/1000 |
| Citeseer | 3,327 | 9,228 | 6 | 3,703 | 120/1500/1000 |
| Pubmed | 19,717 | 88,651 | 3 | 500 | 60/1500/1000 |
| Flickr | 89,250 | 899,756 | 7 | 500 | 44,625/22,312/22,312 |
| Reddit | 232,965 | 114,615,892 | 41 | 602 | 15,3932/23,699/55,334 |
| Ogbn-arXiv | 169,343 | 2,332,486 | 40 | 128 | 90,941/ 29,799/48,603 |
| Amazon-Ratings | 24,492 | 93,050 | 5 | 300 | 50%/25%/25% |
| Actor | 7,600 | 26,659 | 5 | 932 | 60%/20%/20% |

### C.2 BASELINES

In this section, we detailedly introduce existing graph condensation methods that are used as baselines in this paper, including GCond (Jin et al., 2022), GCDM (Liu et al., 2022), SGDD (Yang et al., 2023), and SFGC (Zheng et al., 2023). GCond (Jin et al., 2022), SGDD (Yang et al., 2023), and SFGC (Zheng et al., 2023) are gradient-matching-based methods, whereas GCDM is a distribution-matching-based method.

**Gradient-matching-based methods.** Gradient matching (Zhao et al., 2021) aims to match the network parameters w.r.t. to the large-real and small-synthetic training data by matching the task's gradients at each step. In this way, it wishes the models trained on the original dataset and the synthetic dataset can converge to similar solutions. This is a bi-level optimization problem that can be formulated as following:

$$\min_{\mathcal{S}} \mathbb{E}_{\theta_0 \sim P_{\theta_0}} \left[ \sum_{t=0}^{T-1} D(\theta_t^{\mathcal{S}}, \theta_t^{\mathcal{G}}) \right] \tag{14}$$
$$\text{s.t. } \theta_{t+1}^{\mathcal{S}} = \texttt{opt-alg}_\theta(\mathcal{L}(\text{GNN}_{\theta_t^{\mathcal{S}}}(\mathcal{S}))) \text{ and } \theta_{t+1}^{\mathcal{G}} = \texttt{opt-alg}_\theta(\mathcal{L}(\text{GNN}_{\theta_t^{\mathcal{G}}}(\mathcal{G}))),$$

where $D(\cdot, \cdot)$ is a distance metric measuring the distance between two gradient matrixes. $T$ is the number of steps in the training trajectory. $\texttt{opt-alg}$ is a specific optimization procedure with a fixed number of steps. In other words, the gradient matching algorithm wishes to generate a condensed graph $\mathcal{S}$ such that the GNN parameters trained on them $(\theta_t^{\mathcal{S}})$ are similar to the ones trained on the original training graph $\theta_t^{\mathcal{G}}$.

Since the distance between $\theta_t^{\mathcal{S}}$ and $\theta_t^{\mathcal{G}}$ is usually small during the training process (Zhao et al., 2021; Jin et al., 2022), the above objective can be simplified as follows,

$$\min \mathbb{E}_{\theta_0 \sim P_{\theta_0}} \left[ \sum_0^{T-1} D(\nabla_\theta \mathcal{L}(\mathsf{GNN}_{\theta_t}(\mathcal{S})), \nabla_\theta \mathcal{L}(\mathsf{GNN}_{\theta_t}(\mathcal{G}))) \right] \tag{15}$$

where $\theta_t^{\mathcal{S}}$ and $\theta_t^{\mathcal{G}}$ are replaced by $\theta_t$, which is trained on $\mathcal{S}$. Let $\mathbf{G}^{\mathcal{S}} \in \mathbb{R}^{d_1 \times d_2}$ and $\mathbf{G}^{\mathcal{G}} \in \mathbb{R}^{d_1 \times d_2}$ be the gradient matrixes of a specific layer of $\theta$ on the synthetic graph $\mathcal{S}$ and original graph $\mathcal{G}$, then the distance function for condensation is defined as follows,

$$D(\mathbf{G}^{\mathcal{S}}, \mathbf{G}^{\mathcal{G}}) = \sum_{i=1}^{d_2} \left( 1 - \frac{\mathbf{G}_i^{\mathcal{S}} \cdot \mathbf{G}_i^{\mathcal{G}}}{\|\mathbf{G}_i^{\mathcal{S}}\| \|\mathbf{G}_i^{\mathcal{G}}\|} \right). \tag{16}$$

The above is the standard gradient matching process proposed in (Zhao et al., 2021), which is adopted by GCDM (Liu et al., 2022), SGDD (Yang et al., 2023), and SFGC (Zheng et al., 2023). Then, these methods have distinct designs in how to generate the adjacency matrix $\mathbf{A}'$ of the condensed graph $\mathcal{S}$, and we recommend the readers read the paper for details.

**Distribution-matching-based methods.** Distribution matching (Zhao & Bilen, 2023) learns the condensed graph by directly minimizing the discrepancy (typically the MMD distance) between the distributions of the original graph $\mathcal{G}$ and the synthetic graph $\mathcal{S}$:

$$\min_{\mathcal{S}} \mathsf{MMD}(\mathcal{G}, \mathcal{S}) \tag{17}$$

GCDM defines the distribution of a graph as the set of its multi-hop receptive fields. Formally, $R(i, L)$, denotes node $i$'s $L$-hop receptive fields. For example, $R(1, 1)$ denotes the first-order neighbors of node 1, and $R(1, 2)$ denotes the second-order neighbors of node 1 (including the first-order ones).

According to the definition of Maximum Mean Discrepancy, the class-wise loss function can be formulated as follows:

$$\mathsf{MMD}_c(\mathcal{G}, \mathcal{S}) = \mathsf{sup}_{\phi \in \mathcal{H}} \left\| \frac{1}{|\mathcal{V}_c|} \sum_{i \in \mathcal{V}_c} \phi(R_\mathcal{G}(i, L)) - \frac{1}{|\mathcal{V}_c'|} \sum_{j \in \mathcal{V}_c'} \phi(R_\mathcal{S}(j, L)) \right\|. \tag{18}$$

GCDM treats the above optimization problem as a bi-level optimization problem in the following form:

$$\min_{\mathcal{S}} \max_{\phi} \left\| \frac{1}{|\mathcal{V}_c|} \sum_{i \in \mathcal{V}_c} \phi(R_\mathcal{G}(i, L)) - \frac{1}{|\mathcal{V}_c'|} \sum_{j \in \mathcal{V}_c'} \phi(R_\mathcal{S}(j, L)) \right\|. \tag{19}$$

$\phi$ is parameterized as a graph neural network model which outputs the embedding of a node. Then, the above optimization process involves alternatively updating the parameters of the GNN encoder $\phi$, and the parameters of the synthetic graph $\mathcal{S}$.

The proposed DCGC is also based on distribution matching. Yet, we leveraged the characteristics of MMD in RKHS and kernel tricks to directly optimize in the input space, avoiding the formulated min-max bi-level optimization problem in GCDM that operates in the embedding space. This significantly accelerated the training speed. It also eliminated the reliance on a specific GNN model, resulting in excellent cross-architecture generalization ability.

## C.3 IMPLEMENTATION DETAILS

### C.3.1 CONFIGURATIONS.

We conduct all experiments with:

- Operating System: Ubuntu 22.04.3 LTS
- CPU: Intel 13th Gen Intel(R) Core(TM) i9-13900K
- GPU: NVIDIA GeForce RTX 4090 with 24 GB of Memory
- Software: CUDA 12.2, Python 3.9.16, PyTorch (Paszke et al., 2019) 1.12.1, DGL (Wang et al., 2019) 1.0.1+cu117

### C.3.2    IMPLEMENTATIONS OF INITIALIZING $\mathbf{X}'$ AND $\mathbf{A}'$

We provide the pytorch-style code for initializing $\mathbf{X}'$ and $\mathbf{A}'$ in Algorithm 2.

---

**Algorithm 2:** PyTorch-style code for the initialization of $\mathbf{X}'$ and $\mathbf{A}'$

---

```
# C: number of classes
# Ns: number of nodes in the synthetic graph
# Ns_c: number of nodes from class c in the synthetic graph
# X: node feature matrix of the original graph: N * D
# Y: list of the idxes of training nodes for each class
# Xs: list of node feature matrix of the synthetic graph: [Ns_1 * D, Ns_2 * D, ...,
    Ns_C * D ]
# As: adjacency matrix of the synthetic graph

# alpha_on: factor for initializing the on-diagonal terms of As
# alpha_off: factor for initializing the off-diagonal terms of As

# initialize Xs
for c in range(C):
    Nc = Y[i].shape[0]
    idx = np.arange(Nc)
    np.random.shuffle(idx)
    keep_idx = idx[:Ns_c]

    if Nc >= Ns_c:
        Xc.data[i] = X[Y[i]][keep_idx]
    else:
        Xc.data[i][:Nc] = X[Y[i]][keep_idx]

# initialize As
As = Parameter(torch.ones((Ns, Ns)))
As = As * alpha_off

for i in range(Ns):
    As[i, i] = alpha_on

# force A to be in (0,1)
# on-diag terms being eps_on
# off-diag terms being eps_off
As = torch.nn.functional.sigmoid(As)

return Xs, As
```

---

### C.3.3    IMPLEMENTATIONS OF GRAPH DIFFUSION PROCESS.

We provide the pytorch-style code for the graph diffusion process in Algorithm 3.

---

**Algorithm 3:** PyTorch-style code for the graph diffusion process

---

```
# X: node feature matrix
# Y: node label matrix
# t_A: normalized graph adjacency matrix
# K: number of steps for diffusion

X_list = []
Y_list = []

X_list.append[X]
for k in range(K):
    # feature diffusion
    X = torch.mm(t_A, X)
    X_list.append(X)

    # label diffusion
    Y = torch.mm(t_A, Y)
    y = torch.nn.functional.normalize(Y, p = 1)
    Y_list.append(Y)

return X_list, Y_list
```

---

### C.3.4 IMPLEMENTATIONS OF THE MAXIMUM MEAN DISCREPANCY BETWEEN TWO DISTRIBUTIONS.

We provide the pytorch-style code for computing the MMD loss in Algorithm 4.

---

**Algorithm 4:** PyTorch-style code for the MMD loss

---

```
# X: distribution of source: N1 * D
# Y: distribution of target: N2 * D
# tau: bandwidth

def gaussian_kernel(source, target, tau):
    L2_distance = torch.cdist(source.unsqueeze(0), target.unsqueeze(0)) ** 2

    kernel_val = torch.exp(-L2_distance / tau)
    return kernel_val.squeeze(0)

XX = gaussian_kernel(X, Y, tau)
XY = gaussian_kernel(X, Y, tau)
YX = gaussian_kernel(Y, X, tau)
YY = gaussian_kernel(Y, Y, tau)

return - XY.mean() - YX.mean() + YY.mean() + XX.mean()
```

---

### C.4 ADDITIONAL EMPIRICAL RESULTS

**Impacts of the number of diffusion steps $K$.** The maximum diffusion step $K$ is an important hyperparameter of DCGC, which decides the maximum order of graph structure information to be considered. We here examine DCGC's sensitivity to them. In Fig. 3, we plot the influences of increasing $K$ on DCGC's performance. The observations are summarized as follows: 1) $K = 0$ results in poor performance since it only considers the individual node features without considering the graph structure information. The label distribution information will not be considered by DCGC as well. 2) Increasing $K^f$ can increase DCGC's performance initially, while the performance begins to become stable from $K = 3$. Considering that further increasing $K$ will not improve the performance significantly and will bring additional computational cost, we simply set $K = 3$.

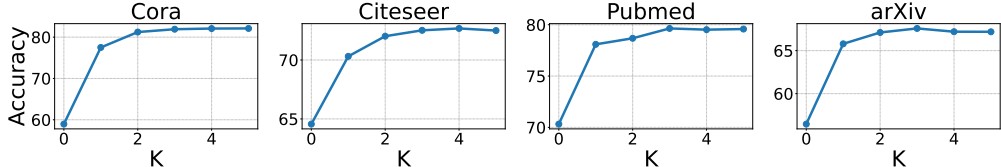

Figure 3: Sensitivity analysis of the number of diffusion steps $K$.

**Impacts of the number of time interval $\Delta t$.** We further study the impacts of the time interval $\Delta t$. As shown in Proposition 1, the diffusion process is stable as long as $0 < \Delta t \leq 1$, while a small $\Delta t$ might help learn more sophisticated evolution of node feature distribution w.r.t. $t$. Therefore, we perform an ablation study on $\Delta t$ with four values $0.25, 0.5, 0.75$, and the default $1$. For a fair comparison, we set the maximum diffusion time as $K \cdot \Delta t = 3$ for all $\Delta t$, which will lead to different maximum steps $K$.

As demonstrated in Fig. 4, while using a short time interval $\Delta t$ usually helps obtain better performance, the improvement is relatively marginal. This demonstrates that the proposed DCGC's performance is not sensitive to $\Delta t$. Considering that the overall training time is linear w.r.t. $K$, we prefer to select the maximum possible time interval $\Delta t = 1$.

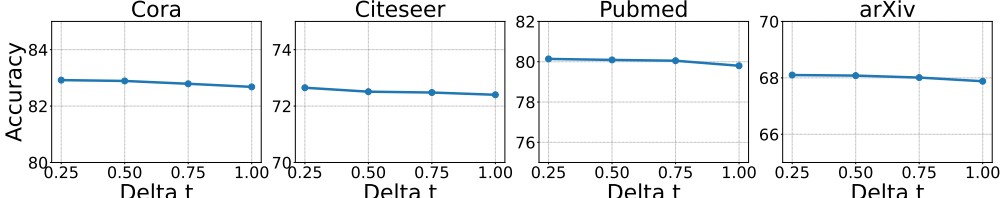

Figure 4: Sensitivity analysis of the time interval $\Delta t$.

