# OpenReview forum: "Data-Centric Graph Condensation via Diffusion Trajectory Matching"
_ICLR.cc/2025/Conference — ICLR 2025 Conference Withdrawn Submission_

### Official Review · Reviewer_Umtp · 2024-10-28

**Soundness:** 3
**Presentation:** 2
**Contribution:** 3
**Rating:** 5
**Confidence:** 3

**Summary:**

This paper proposes DCGC, a graph condensation method. DCGC defines the distribution of a graph as the trajectories of its node signals induced by a diffusion process over the geometric structure. Two advantages of DCGC is efficiency and cross-model generalizability.

**Strengths:**

S1: The efficiency of graph condensation is an important problem in this field.

S2: This paper is easy-to-follow and the derivation is clear.

S3: Experiments are standard, the authors seem to have a good understanding of this field.

**Weaknesses:**

W1: It would be better to include recent baselines, I have seen many works in efficient graph condensation this year.

W2: There are no experimental results with large graphs, but I understand that previous work didn't include large graphs either.

W3: The presentation could be improved a bit. Figure 1 looks somewhat rough, and the standard deviation for SFGC on cora is missing in Table 1.

**Questions:**

Q1: About W1, I recommend add GDEM [1], GCSR [2], Exgc[3], GC-SNTK [4], GEOM [5]. Efficiency is more important than accuracy. I think Exgc and GC-SNTK is about efficiency of graph condensation (I am not very sure, the authors could double-check), thus they are highly related to this paper.

[1] Graph distillation with eigenbasis matching

[2] Graph data condensation via self-expressive graph structure reconstruction.

[3] Exgc: Bridging efficiency and explainability in graph condensation

[4] Fast Graph Condensation with Structure-based Neural Tangent

[5] Navigating Complexity: Toward Lossless Graph Condensation via Expanding Window Matching

Q2: About W2, I think Ogbn-products could be included.

---

### Official Review · Reviewer_4VqV · 2024-10-30

**Soundness:** 2
**Presentation:** 2
**Contribution:** 3
**Rating:** 5
**Confidence:** 3

**Summary:**

The paper introduces a method for graph condensation, which aims to create a smaller graph that approximates a larger target graph. The method begins with a small dense graph, optimizing its edge weights and node features using a Graph Neural Network (GNN) to minimize the Maximum Mean Discrepancy (MMD) between the diffusion trajectories of node features and labels in both the small and target graphs. The method is evaluated on node classification benchmarks, demonstrating state-of-the-art accuracy, albeit with marginal improvements, and significantly faster training times compared to previous graph condensation methods.

**Strengths:**

The use of diffusion trajectory distributions to capture the intertwined structural and feature information of a graph, and employing MMD as a proxy for graph similarity in graph condensation, is an interesting and novel idea.

The method offers a substantial reduction in training time, which is a significant advantage over existing graph condensation techniques.

**Weaknesses:**

**Clarity Issues**:

i) The overall objective of the method is not clearly defined. Section 3.1 (Problem Formulation) begins by defining Graph Condensation via Distribution Matching, but the general objective of graph condensation is not explicitly stated. From previous work, it appears that the goal is to condense a large graph into a smaller one, enabling the performance of a node label classifier trained on the condensed graph to be transferred to the target graph. Is this the most general definition of graph condensation? I recommend that the authors clearly and formally define the objective of graph condensation and clarify how the optimization objective proposed in the paper relates to this general objective. Is it a proxy? A specific instance?

ii) In the introduction of the methodological section, the distinction between node features and labels is not clarified. It is only from previous work that one can infer that the node labels are the target for the classifier when evaluating the condensed graph.

iii) The evaluation process is not clearly described. The paper merely states that a GCN is trained on the condensed graphs and accuracy is reported on the "testing" nodes of the original graph. What are the testing nodes? Are they present or absent during the optimization of the condensed graph? Which accuracy is being referred to? In Table 6, numbers are reported for training/validation/testing. What do these numbers represent? What is the distinction between testing and validation? There is no prior mention of a validation set in addition to the testing set.

iv) The datasets used in the experiments are not described. What do the graphs represent? What are the node features and labels? I encourage the authors to provide a description of the datasets used in the experiments, in addition to reporting dataset statistics.


**Complexity Analysis**: The complexity analysis is not well justified. The paper states the asymptotic complexity for different stages of the method without justification. I suggest providing a brief explanation of how these complexities are derived.


**Visualization**: The paper lacks visualizations of the generated condensed graphs. It would be helpful to see examples of both original and condensed graphs to better understand the method.


**Overall Benefits**: The overall gain of graph condensation is not clear from the experimental results. It would be insightful to report the training time of the classifier on both the condensed graph and the target graph. This would help determine if there is an overall gain in training time, i.e., if synthesizing the condensed graph and training the classifier on it is faster than training the classifier directly on the target graph.

**Questions:**

see weaknesses

---

### Official Review · Reviewer_u722 · 2024-11-03

**Soundness:** 2
**Presentation:** 2
**Contribution:** 1
**Rating:** 3
**Confidence:** 5

**Summary:**

The authors proposed Data-Centric Graph Condensation (DCGC), a approach to condensing large graphs by aligning the distribution between the original and synthetic graphs. DCGC defines a graph's distribution through node signal trajectories, influenced by a diffusion process that captures multi-order (k-hop) structural information using node features and labels.

**Strengths:**

DCGC doesn’t require training of the original graph like the existing work for graph condensation.
Model agnostic and independent from any specific set of hyper-parameters.

**Weaknesses:**

Novelty: The idea seems to be a simple generalization of the work (Zhao & Bilen, 2023) to the graph data. (Equation 2, 4 from Zhao & Bilen, 2023 ~ Equation 1, 2).
In Equation 1, the objective is defined as matching the distribution of nodes that belong to the same class (class-conditional set defined in line 130), while in real-world graphs, edges between different nodes of different classes may exist; ignoring this fact into the objective may result in a low-quality condensed graph from some graphs.
Presentation:
(a) Some of the content looks redundant. Example: Definition 1 and Definition 2 can be merged as they summarize the same idea.
(b) Better include a subsection to declare heterophilic graphs, as understanding the heterophilic idea from Figure 1 b) can be misleading.
c) In line 248 please define E, D and C.
Missing recent graph coarsening work as baseline [1], which are by default models and hyperparameters agnostic.
Reproducibility : Authors should include the code for better reproducibility rather than figures of code (Figure 2 and Figure 3)

**Questions:**

Line 41: “However, these methods often rely on some predefined heuristics and lack guidance from training…." Is the sole goal of graph condensation to produce a condensed graph that performs well exclusively on GNN models? If not, does this imply that graph coarsening is a more advantageous option when the downstream task is not to train a GNN but rather something different?
Can authors comment on any other metric other than GNN model accuracies to quantify the quality of condensed graphs?
Line 284: for large and dense graphs, this computational cost can be large.
Line 317: N’ = N x r. In table 1, r values are greater than 1 for the cora dataset, and as N’ << N there is some inconsistency.
Can we use the condensed graph for other downstream tasks? (Example: train a model using condensed graph to do graph clustering.)
May the authors incorporate current efforts on graph coarsening and condensation [1,2] into the baseline and results?

[1] Kumar, Manoj, et al. "Featured graph coarsening with similarity guarantees." International Conference on Machine Learning. PMLR, 2023.
[2] Kumar, Manoj, et al. "Optimization Framework for Semi-supervised Attributed Graph Coarsening." The 40th Conference on Uncertainty in Artificial Intelligence.

[3] Gupta, Mridul, et al. "Mirage: Model-agnostic graph distillation for graph classification." arXiv preprint arXiv:2310.09486 (2023).

---

### Official Review · Reviewer_gipW · 2024-11-08

**Soundness:** 3
**Presentation:** 3
**Contribution:** 3
**Rating:** 6
**Confidence:** 4

**Summary:**

The paper presents a technique for graph condensation, where we want to reduce the size of a graph, but keep the main characteristic of the graph intact. The approach works by defining the distribution of a graph as the trajectories of its node signals induced by a diffusion process over the geometric structure. Using the assumption, the technique generates a much smaller graph by aligning the distribution of generated and original graphs in this diffusion process. The authors then show the benefit of the proposed approach in theory and practice.

**Strengths:**

1. Very interesting idea on using diffusion process for graph condensation task
2. Unlike many previous works, the approach does not rely on the information from a trained GNN on the full graph.
3. The (unsupervised) training procedure is solely based on the data (graph input), and does not depend on what GNN architecture is going to be used.
4. The approach also considers the node label distribution within the graph structure, to preserve homophily/heterophily property in the generated graph.
5. The paper provided detailed explanations on the motivations, theoretical grounding, algorithm, training procedure, and experiments.
6. The experiment results demonstrate the benefit of the proposed technique.

**Weaknesses:**

1. It will be good to provide a more thorough analysis on the case where the architecture-dependent method such as SFGC may perform better than the proposed approach, and vice-versa. Is SFGC performing better when certain architecture is used, or a certain condensation ratio is used, or totally depends on the dataset.
2. In the heterophilic graph experiments, it will be good to have other GNN architectures as well, such as the ones used in the standard experiments.
3. It will be good to have results on more condensation ratios, to see how the methods stack up on various settings.
4. Some of the chosen condensation ratios in the experiments look a bit random (e.g. 2.6%, 1.8%, 3.6%). Could the authors explain the decision to use the ratios?

**Questions:**

Please address the weaknesses mentioned above.

---

### Note · Authors · 2024-12-03

**Comment:**

We thank all reviewers for their review, which made us realize that the current manuscript still needs improvement in many areas. We will consider your comments and suggestions.

**Withdrawal Confirmation:**

I have read and agree with the venue's withdrawal policy on behalf of myself and my co-authors.